# Time Series Classification of Autism Spectrum Disorder Using the Light-Adapted Electroretinogram

**DOI:** 10.3390/bioengineering12090951

**Published:** 2025-09-02

**Authors:** Sergey Chistiakov, Anton Dolganov, Paul A. Constable, Aleksei Zhdanov, Mikhail Kulyabin, Dorothy A. Thompson, Irene O. Lee, Faisal Albasu, Vasilii Borisov, Mikhail Ronkin

**Affiliations:** 1Engineering School of Information Technologies, Telecommunications and Control Systems, Ural Federal University named after the First President of Russia B. N. Yeltsin, Yekaterinburg 620002, Russia; wumyup@yandex.ru (S.C.); anton.dolganov@urfu.ru (A.D.); falbasu@urfu.ru (F.A.); v.i.borisov@urfu.ru (V.B.); 2Caring Futures Institute, College of Nursing and Health Sciences, Flinders University, Adelaide, SA 5042, Australia; paul.constable@flinders.edu.au; 3VisioMed.AI, Moscow 125212, Russia; kulyabin@visiomed.ai; 4Tony Kriss Visual Electrophysiology Unit, Clinical and Academic Department of Ophthalmology, UCL Great Ormond Street Institute of Child Health, Great Ormond Street Hospital for Children NHS Trust, The University College London, London WC1E 6BT, UK; dorothy.thompson@ucl.ac.uk; 5UCL Great Ormond Street Institute of Child Health, University College London, London WC1E 6BT, UK; 6Population Policy and Practice Programme, Behavioural and Brain Sciences Unit, UCL Great Ormond Street Institute of Child Health, University College London, London WC1E 6BT, UK; irene.lee@ucl.ac.uk

**Keywords:** neurodevelopment, retina, electroretinogram, waveform, explainable AI, time-series classification

## Abstract

The clinical electroretinogram (ERG) is a non-invasive diagnostic test used to assess the functional state of the retina by recording changes in the bioelectric potential following brief flashes of light. The recorded ERG waveform offers ways for diagnosing both retinal dystrophies and neurological disorders such as autism spectrum disorder (ASD), attention deficit hyperactivity disorder (ADHD), and Parkinson’s disease. In this study, different time-series-based machine learning methods were used to classify ERG signals from ASD and typically developing individuals with the aim of interpreting the decisions made by the models to understand the classification process made by the models. Among the time-series classification (TSC) algorithms, the Random Convolutional Kernel Transform (ROCKET) algorithm showed the most accurate results with the fewest number of predictive errors. For the interpretation analysis of the model predictions, the SHapley Additive exPlanations (SHAP) algorithm was applied to each of the models’ predictions, with the ROCKET and KNeighborsTimeSeriesClassifier (TS-KNN) algorithms showing more suitability for ASD classification as they provided better-defined explanations by discarding the uninformative non-physiological part of the ERG waveform baseline signal and focused on the time regions incorporating the clinically significant a- and b-waves of the ERG. With the potential broadening scope of practice for visual electrophysiology within neurological disorders, TSC may support the identification of important regions in the ERG time series to support the classification of neurological disorders and potential retinal diseases.

## 1. Introduction

The clinical electroretinogram (ERG) is an important non-invasive diagnostic test used in ophthalmology to assess the functional state of the retina by recording the changes in the bioelectrical potential obtained following brief flashes of light. The shape of the recorded ERG signal or “waveform” varies depending on the state of light or dark adaptation of the retina and the stimulating flash parameters (duration, strength, frequency, and spectral composition) [1]. Interpretation of the ERG waveform supports diagnosis in retinal conditions [2] and disorders affecting the broader central nervous system [3]. One potential use of the ERG is in the classification of neurodevelopmental disorders such as autism spectrum disorder (ASD) [4,5,6].

The light-adapted ERG signal consists of two main components, designated the a- and b-waves, shown in Figure 1. The a-wave is generated by the photoreceptors that hyperpolarize following light onset, generating a negative deflection in the signal [7]. The light-adapted b-wave is generated by the post-receptoral inner cells of the retina (bipolar, amacrine, ganglion, and glial) with the summated combination of hyperpolarizing and depolarizing currents from these cells contributing to the overall shape and amplitude of the positive b-wave [8,9,10]. The Oscillatory Potentials (OPs) are high-frequency components visible on the rising limb of the b-wave and originate in the amacrine cells [11]. The main time domain features that are considered for clinical diagnoses are the amplitudes and time to peaks of the principle a- and b-waves. In Figure 1, these parameters are denoted as (Va, Ta) for the a-wave and (Vb, Tb) for the b-wave [1]. In ASD, the prominence of the OPs has been reported to be atypical in adults [12] and younger cohorts [4]. In some cases, other components of the ERG signal can also be considered [13]. Typically, the duration of the clinical components of the ERG waveform ranges from 100 to approximately 250 ms, depending on the clinical test, and the frequency range of the signal lies between 0 and 300 Hz [14]. However, the main informative characteristics of the signal may differ from the protocols of recording for different clinical applications [15].

Figure 1 depicts light-adapted ERG waveforms with the negative a-wave and positive b-wave labeled with the flash stimulus onset indicated by the white arrow. The amplitude on the a-wave Va is measured from the baseline to the trough of the a-wave, and the amplitude of the b-wave (Vb) is measured from the a-wave minima to the b-wave maxima. The time to peaks is taken from the stimulus onset to the peak of the a- and b-waves (Ta and Tb), respectively.

ERG waveform analysis is generally based on the four aforementioned time-domain features, which are typically extracted by a combination of manual inspection and in-built algorithms to find the peaks of the principle a- and b-waves [1,16]. The high-frequency OPs contained within the 75–300 Hz range are generally reported qualitatively and/or quantitatively as the sum of the peaks, or individual peaks of interest [1]. OFF-line packages are also available for analyzing the ERG signal into these time-domain components [17].

However, the specificity of these fixed time point features do not fully capture the dynamic patterns of change in the ERG waveform. Developing novel approaches to ERG signal analysis would allow the enhancement of the ERG response descriptions and, thus, improve the precision of the analysis to expand the clinical utility of the ERG [18]. Previous studies have implemented Fourier spectrum-based analysis, time-frequency methods including short-time Fourier transform, and continuous wavelet transform and discrete wavelets [16] that use a 2D power spectrum density or scalograms, corresponding to the signal representation methods. The features of these representations can be extracted either manually using classical machine learning algorithms [13] or automatically using 2D convolutional neural networks [19]. It should be noted that approaches that use the 1D neural network for ERG signal classification [20] that uses non-linear decision-making algorithms can suffer from a lack of explainability [13,19]. To evaluate the shape or trajectory of the ERG time series, an ideal model would extract important features that have a clinical interpretation. To address this possibility, Time-Series Classification (TSC) methods for signal classification provide native and accurate methods that can also be applied to the ERG signal [21]. The advantages of these methods have been demonstrated in well-known univariate [22] and multivariate datasets [23] as well as in several healthcare [24] and industrial applications [25]. Most of the works in TSC do not include explainability analysis issues, although some general attempts have recently been made [26,27,28,29].

From an application point of view, the explainability of machine learning algorithms plays a critical role in medical decision support systems. The explainability shows the logic of automatic decision-making systems, which directly affects the trustworthiness of the system and increases the feeling of safety for patients. It also helps physicians to better understand patient data and, on a deeper level, provides insights into the features that affect the particular model’s decision making that might not be apparent during manual feature extraction.

Recent studies have increasingly applied machine learning and advanced signal processing in addition to traditional methods to ERG for neurodevelopmental and ophthalmic diagnostics, including ASD, ADHD, and glaucoma, reflecting a growing interest in data-driven approaches to retinal electrophysiology [6,30,31,32]. Additionally, several works have demonstrated the potential of ERG-based phenotyping and time-series modeling for clinical applications, supported by improvements in explainability, reproducibility, and integration with psychometric or structural biomarkers [33].

Decision-making algorithms should take into account relevant clinical factors instead of random correlations or biases in the data [34]. Explainability methods should be intuitively understandable for health care practitioners and visually demonstrate the importance of the data parts. Hence, of the evaluated explainable artificial intelligence methods, SHapley Additive exPlanations (SHAP) appears to be most suited for capturing the time points relevant for classification behavior [35]. In the case of TSC, explainability not only shows the points of potential interest, but also highlights important time-dependent intervals (and their influence on decisions) that provide more information for the clinician to analyze the algorithm’s result in the appropriate clinical context [36,37]. As an example, for ASD, model explainability could be useful in identifying which regions of the ERG signal influence the final prediction and which of these are affected the most/least in individuals with ASD compared with typically developing individuals or other neurological disorders [3].

This paper illustrates the use of time-series-based machine learning methods to classify ERG signals, with the aim of interpreting the decisions made by the models in order to better understand the classification process. In this instance, the dataset comprises two classes, ASD and control, but TSC could be used in any classification of disorders affecting the ERG signal, whether they are neurological or retinal in origin. This work provides the following contributions to the field of signal analysis:To the best of the authors’ knowledge, this is the first application of SHAP explanation to the TSC algorithms’ results for an ERG signal classification task.SHAP methods were applied on the TSC models to provide a domain-agnostic explanation, highlighting the important regions of the signals for classification.

The results were limited to the default implementation of machine learning TSC methods, which are found in related frameworks; see, for instance, sktime [38]. The results obtained here using TSC could be considered as a baseline for further investigations in ERG classification. SHAP implementation was also given in the original authors’ code implementation [39].

Therefore, the aim of this study was to apply and evaluate TSC algorithms, combined with SHAP-based explainability, to classify light-adapted ERG signals from individuals with ASD and control participants, as well as to identify the most informative regions of the ERG waveform contributing to the classification.

This paper is structured as follows: Section 2 discusses the data used for the study and its characteristics as well as the time series models the data were trained on and the implementation details including data splitting for training and testing, hyperparameter tuning, and SHAP implementation for the model’s explainability. Section 3 presents the results obtained, model evaluation, and analysis of implemented algorithms and discusses observations and insights derived from the models’ results as well as the explainability methods applied. Section 4 draws conclusion based on obtained results and insights derived from the models and explainability techniques and what that means for the current and future research.

## 2. Materials and Methods

### 2.1. Dataset

The dataset under consideration contained a total of 991 ERG light-adapted signals taken from 30 ASD and 20 typically developing control participants [20,40]. The dataset contained the following data: ID of the person, group (control or ASD), eye (left or right), flash strength (1.204, 1.114, 0.949, 0.799, 0.602, 0.398, 0.114, −0.119, −0.367) log cd.s.m^−2^ denoting the various flash strengths used during the signal recording sessions. For further details on the participants and recording methods, see prior works [6]. For this particular study, the signal strength parameter 1.204 log cd.s.m^−2^ was used because this strength has previously been shown to be a strong discriminator of the ASD and control groups [4,5]. In total, there were 153 unique IDs that satisfied such criteria, among them there were 88 control and 65 ASD signals. Before the machine learning procedure, the data were randomly split into train and test subsets with a proportion of 80 to 20, stratifying by the target column. The split was performed at the individual level so that the signals from the eyes of one individual could only be in the train or the test subsets. Typical signals for different flash strengths are shown in Figure 2. Visual analysis of Figure 2 shows a reduced positive b-wave amplitude in the ASD cases.

### 2.2. Time Series Classification Models

The following machine learning algorithms for TSC were implemented in the sktime library [38] coupled with the explainability methods to provide high accuracy and transparency in the decision-making process [27].

Word extraction for TSC (WEASEL) is a dictionary-based classifier for time-series data. The algorithm is based on the bag-of-patterns representation, which consists of extracting sub-sequences of different lengths from the time series, discretizing each sub-sequence into a coarsely discrete-valued word, then building a histogram from word counts and training a logistic regression classifier on this bag [41]. In the WEASEL case, it was expected that explainability allowed for highlighting which symbolic patterns (e.g., specific wave chunks) or sub-sequences contributed most to class classification.Time Series Forest (TSF) is an ensemble method for TSC that builds multiple decision trees. The algorithm selects each tree in the forest through random selection of several intervals with randomized lengths and positional offsets. For each sampled interval, three statistical features are computed: the mean, the standard deviation, and the slope. The features of each interval are then aggregated into a composite feature vector that subsequently serves as the input feature space for the construction of a decision tree. The resulting trees are then integrated into the ensemble model. The random forest-like classifier algorithm is applied to all trees [42]. In the TSF case, it was expected that explainability would allow the identification of the critical intervals that corresponded to a clinically relevant interpretation of ERG signals for classification.KNeighborsTimeSeriesClassifier (TS-KNN) is an implementation of the k-nearest neighbors algorithm specifically designed for time series with Dynamic Time Wrapping Distance (DTW). DTW is an elastic distance measure that optimally compares two sequences by warping them non-linearly in time. DTW was applied instead of the traditional Euclidean distance due to the robustness of signal delays (latency) and other distortions in the time domain [43]. In the TS-KNN case, it was expected that explainability would allow for the identification of waveform intervals that corresponded to important patterns of behavior in the ERG signal for classification.Random Convolutional Kernel Transform (ROCKET) is a method for TSC based on 1D convolution kernels with random parameter selection. It works by generating a large variety of kernels, each containing different parameters (length, weight, dilation, etc.), and applies these kernels to the data through convolution. Each convolution results in two features, the positive value amount and the maximum value [44]. In this work a random forest was applied to the vector of features obtained for 3000 kernels. In the ROCKET case, it was expected that explainability would identify regions and points (e.g., a-wave amplitude) emphasized by the kernel emphasis, which correspond to differences in retinal signaling between the groups for classification.

### 2.3. Hyperparameter Selection

The selection of hyperparameters in machine learning algorithms is a key step that can significantly impact the performance of the model. Hyperparameters are parameters that are not directly learned by the model, but are set in advance. Choosing these parameters correctly can improve accuracy, avoid overfitting, and improve the generalization ability of the model [45]. In this case, the Grid Search method was used, which involved testing combinations of hyperparameters from given ranges. Conceptually, this process consisted of a complete code execution cycle with an equal number of random_state values. To ensure robustness of the evaluations, the repeated hold-out split (split into training and test samples) was used at each step of the cycle.

At each stage of the cycle, the following conditions were applied:If there was no variation in hyperparameters in the algorithm, then it was run once and the classification metrics values were saved to memory.If the algorithm had variations of hyperparameters, then it was run in the amount of these variations and all of the obtained classification metrics were saved to memory.

After going through all possible variations of random_state in the loop, the mean, median, and standard deviation were calculated for each model hyperparameter. The list of the parameter values used is as follows:random_state —112, 1231, 42, 990, 2500, 467, 777, 89, 258, 24;n_estimators of the TSF algorithm—10, 50, 100, 200, 300, 400, 500;n_neighbors of the TS-KNN algorithm—1, 2, 3, 4, 5, 6, 7;num_kernels of the ROCKET algorithm—100, 1000, 10,000, 20,000, 30,000.

There were no hyperparameters for the WEASEL algorithm. The list of parameters obtained from the algorithms used and for which the mean, median, and standard deviation were calculated is as follows:F1-score (separately for control and ASD individuals;Balanced accuracy;Time for training.

### 2.4. Explaining TSC Models Using the SHAP Library

The SHAP (SHapley Additive exPlanations) library is a powerful tool for explaining predictions of machine learning algorithms. The main idea of SHAP is to use game theory to determine the contribution of each feature to the model’s predictions. The advantages of SHAP include high accuracy of interpretation, applicability to various models (e.g., decision trees, regression models) and convenient visualizations for feature analysis. Limitations of SHAP are related to the computationally expensive process, especially for complex models and large datasets [39].

SHAP does not directly support all types of models created with sktime, but it is possible to use SHAP with models that support the scikit-learn interface, such as models that implement the *.fit()* and *.predict()* methods. The algorithms used in this paper met this requirement. The general approach is summarized as follows:Train a time-series model using sktime.Use the trained model to make predictions on the test data.Apply SHAP to the model to obtain feature importance values.

As sktime models use the whole signal as the input, the SHAP library can be used to test which part of the signal contributes to the model output. This then shows where the SHAP algorithm decomposes prediction for a particular signal as a linear combination of its time steps [39].

## 3. Results

### 3.1. Visual Inspection of the Signals

An average signal in the context of data analysis using machine learning algorithms is often used to identify common patterns in data grouped by certain conditions or categories. Visualization of a signal average can be presented in the form of the following plots, which help to visually identify trends, anomalies, or differences between conditions. An average signal for all control and all ASD participants was constructed from the original data and is shown in Figure 3. The upper subplots represent the averaged signals for each group (line indicates the mean value of the signal and surrounding area indicates ± one standard deviation). The lower subplots show the individual signals for each group. Previous studies using this dataset have highlighted the leading edge of the b-wave that contains the OPs as being important for ASD classification [46]. Consequently, the time interval containing the b-wave was given more importance for feature selection.

Visual analysis of Figure 3 led to the following conclusions: the first 20 ms of the signals were highly uninformative because this region represents the baseline or pre-stimulus region. The interval containing the main response from 25 to 60 ms contained the most likely interval with which features could be identified to classify the groups.

### 3.2. Model Evaluation

The machine learning TSC models WEASEL, TSF, TS-KNN, and ROCKET were tested on the dataset described in Section 2.1. The models’ hyperparameters named in Section 2.3 were selected as follows. The Grid Search of the best hyperparameters was based on a 10× repeated train–test split with different random states.

After analyzing the obtained results, the best hyperparameters were selected as follows:n_estimators of the TSF algorithm was 10.n_neighbors of the TS-KNN algorithm was 3.num_kernels of the ROCKET algorithm was 20,000.

Table 1 shows a summary results for the TSC models with the best hyperparameters for the test data. The results shown in the table are mean ± and one standard deviation (STD) for each algorithm.

Table 1 shows that the best results were obtained with the ROCKET algorithm, although the STD was the greatest. This was likely due to the very random nature of ROCKET that may have been improved through ensembling or increasing the number of ROCKET kernels to stabilize the results. The current ROCKET implementation was a trade-off between reliability and complexity of the computation. The worst results were obtained with the WEASEL algorithm and were likely the result of the coarse nature of the dictionary-based ERG signal description, which was not fine enough for the task.

### 3.3. Analysis of Algorithm Errors

Error analysis in machine learning algorithms is an important step in the process of developing models, which helps to improve their quality and make more robust conclusions. The methods .predict() and .predict_proba() were used to obtain predictions from the model, but each of them did it differently, which affected the approaches to error analysis. The .predict() method returned the classes that objects belonged to based on the trained model. For classification problems, this method is often used to determine the final prediction. The .predict_proba() method returns the probabilities of objects belonging to each class. This is useful if additional information about the uncertainty of the predictions is required. For each algorithm, we used .predict() and .predict_proba() to obtain the model’s prediction and prediction probabilities, respectively; then we compared them with the y_test values that were calculated.

When analyzing the obtained results, the ROCKET algorithm was selected as the one having the fewest mistakes from the models that were tested.

Before plotting the average signal of the original data, a condition was added for dividing it into subgroups by comparing the values of y_train and y_pred for each value of the array, as follows:If y_train was equal to 1 and y_pred was equal to 0, then mark the array element as Falsely ASD (or false positive).If y_train was 0 and y_pred was 1, then mark the array element as Falsely control (or false negative).If y_train was 0 and y_pred was 0, then mark the array element Correct control (or True Negative).If y_train was equal to 1 and y_pred was equal to 1, then mark the array element Correct ASD (or true positive).

As a result, four data arrays were obtained for the best-performing algorithm (ROCKET), which were then visualized using the same technique previously described in Figure 4 and Figure 5.

Figure 4 and Figure 5 analysis shows some common patterns: generally the shape of signals, classified as true-positive and true-negative ones, each having three local amplitude peaks at 40 ms, 45 ms, and 55 ms. What is different between these two groups was the lower amplitude of the b-wave peak at 55 ms for the true-positive signals. False-negative signals had the common factor of not having a pronounced peak at 40 ms. The common factor for the false-positive signals was a high amplitude of the b-wave peak at 55 ms.

### 3.4. Explanation of the Signals

The explanation of the model classification results was made with the SHAP algorithm. The plotted results for WEASEL, TSF, TS-KNN, and ROCKET classifiers are shown in Appendix A following the logic: Figure A1, Figure A2, Figure A4, Figure A5, Figure A7, Figure A8, Figure A10 and Figure A11, and show the plots of four signals with the SHAP explanation given from true-positive, false-positive, false-negative and true-negative results obtained by the corresponding TSC, where the positive class corresponds to the ASD group and the negative class corresponds to the control group. The colors indicate the relation of the time step to either positive (ASD group) or negative (control group) class: the red marker implied that the time step makes a contribution to the positive class prediction (ASD); the blue marker implied that the time step makes a contribution to the negative class prediction (control). The white marker indicated that the time-step made no significant contribution to either of of the classes. Additionally, the size of the marker is related to the absolute value of the SHAP coefficients, which helps to identify the regions of greatest importance for classification in the ERG time series by visual inspection.

Additionally, Figure A3, Figure A6, Figure A9, and Figure A12 display the summary of TSC explanations with SHAP for the positive and negative classes for different signals.The positive values on the SHAP plots are associated with time steps that increase the probability that the model will make a correct prediction for ASD. Meanwhile, the negative values are associated with time steps that decrease the probability that the model will make a correct prediction for ASD.

Signal analysis with SHAP allowed for the following conclusions regarding the algorithm explanations.

ROCKET results (see Figure A1, Figure A2 and Figure A3). The explanations of this algorithm were the most sparse and not spread along the whole time series. Similar to the TS-KNN algorithm, the ROCKET algorithm largely ignored the initial baseline part of the signal. The most indicative parts for the predication were associated with the 35 ms and 45 ms time steps, as well as spread throughout the final part of the signal.TS-KNN results (see Figure A4, Figure A5 and Figure A6). The first 25 ms of the signal was mostly ignored by the algorithm. The most significant parts of the signals for positive class prediction were associated with the 35–45 ms interval, as well as with the end part of the signal. The significant part for negative class prediction was mostly located in the 35–45 ms region.WEASEL results (see Figure A7, Figure A8 and Figure A9). Throughout the entire signal, there were significant deviations for both control and ASD individuals. The SHAP coefficients were associated with the first part of the signal baseline (from 0 to 25 ms), as well as significant contribution of the middle part (around 50 ms), and the end of the signal (around 85 ms and 100 ms). In general, the WEASEL algorithm failed to highlight the significant parts of the signal associated with the b-wave.TSF results (see Figure A10, Figure A11 and Figure A12). Similar to the WEASEL algorithm throughout the entire signal, there were significant deviations for both control and ASD individuals. The most significant signals parts associated with importance for classifications were related to the 50 ms, 60 ms, 75 ms, and 100 ms marks. Overall, the TSF algorithm used the whole signal for prediction but failed to identify specific local significant regions of the signal.

Overall, the TS-KNN and ROCKET algorithms were most suitable for predicting or classifying the ASD and control groups based on the central and end parts of the signal. In contrast, WEASEL and TSF algorithms were not suitable for accurately predicting or classifying between the ASD and control groups, because they failed to find local informative parts of the signal.

### 3.5. Discussion

The application of the TSC approach to visualizing important regions of the ERG time series that are important for classification offers a new approach to improve a machine learning model for disease classification with the ERG. Future applications may include retinal disorders such as Congenital Stationary Night Blindness for phenotypic classification [47,48], glaucoma [49,50], neurodevelopmental [5,6,51], and neurodegenerative disorders [52,53,54,55,56]. The advantage of the TSC modeling approach with SHAP provides the clinician with a clear indication of not only the location but the magnitude of the contribution each region makes to the classification of the class.

In the cases presented in this analysis, the TS-KNN and ROCKET algorithms both correctly ignored the uninformative non-physiological part of the signal baseline from 0 to 25 ms. The ROCKET algorithm had the highest Balanced Accuracy of 0.66 with an SDV of 0.10. The performance was lower than previous analyses that used a larger dataset with signal analysis (Balanced Accuracy of 0.88) for male ASD participants [6]. With one flash strength and one eye, a higher Balanced Accuracy of 0.70 was achieved with a random forest classifier [57]. A Balanced Accuracy of 0.81 was achieved using variable frequency complex demodulation of the ERG signal [58]. Despite the lower Balanced Accuracy for classification, the additional information provided by the SHAP analysis of the series provides a clearer clinical picture of which parts of the ERG were the most important for class classification from the algorithms.

For the case of ROCKET, the red markers indicated regions of the ERG waveform time series that contributed to the true ASD classification. These were localized to the a-wave, the second prominent OP peak, and the descending portion of the b-wave. These regions of the ERG waveform have previously been identified as markers for ASD classification. The reduced a-wave has been noted previously [5] as with the OPs and the amplitude of the b-wave [4,5], but the descending portion of the b-wave has not been highlighted previously and may indicate subtle changes in the off-bipolar and ganglion cell contributions to the later part of the signal that were not evident in traditional time domain markers using the photopic negative response [51]. Because the neural generators of the ERG have been described with the light-adapted a-wave shaped by the cone photoreceptors and postsynaptic OFF-pathway neurons with the b-wave formed and shaped by bipolar, amacrine, glial and ganglion cells [8], it is possible to infer that the neurotransmitters involved including glutamate, GABA, and dopamine may be contributing to the altered signaling and therefore the shape of the ERG in this ASD population. Future studies may attempt to investigate genotype–phenotype correlations with TSC parameters to provide greater insights.

TSC with SHAP enables a new perspective on where and to what extent regions of the ERG signal contribute to the classification of disorders. This approach has not been incorporated previously in visual electrophysiology where clinical interpretation of waveforms is an important aspect of diagnosis and classification [2]. With the potential broadening of the scope of practice for visual electrophysiology within neurological disorders, both TSC and ROCKETmay provide key regions to be identified and linked with different neurological disorders ranging from schizophrenia to Parkinson’s disease [3,18,59].

In practice, the use of TSC provides an important new methodological approach that helps to identify local regions of interest that contribute to group classification contained within the ERG time series signal. Although machine learning algorithms and deep learning models have been applied to this dataset previously, with SHAP analysis [6,57], the additional information provided by TSC and SHAP with respect to local features of importance was not evident. In clinical cases, identifying local subtle variations in the ERG waveform could support early identification of retinal disorders. Other statistical approaches that remodel the ERG signal may also provide supporting methods with which to identify regions that are correlated or have an altered trajectory with respect to the time of the registered time series [46]. The application of TSC to the classification of ERG signals builds on these previous studies that have applied signal analysis and statistical methods to develop robust classification models to exploit the ERG’s potential [18] to classify retinal and CNS disorders [3].

The study [60] provides compelling evidence that altered retinal function, as measured by ERG, is present not only in individuals with ASD but also in their unaffected family members, particularly mothers and siblings. These findings suggest that ERG may serve as a potential early biomarker for ASD risk, even before behavioral symptoms emerge. Since motion coherence deficits and ERG alterations follow a familial pattern, ERG could be considered for screening in genetically predisposed populations. More longitudinal and genetic studies are needed to validate the predictive utility of ERGs in early ASD detection [60].

## 4. Conclusions

This study demonstrates the feasibility of applying well-known TSC techniques—WEASEL, TS-KNN, ROCKET, and TSF—to the classification of ERG signals. To our knowledge, this represents an initial step towards adapting these methods for ERG-based diagnostics. The default implementation of the analyzed TSC methods showed less accurate results than Deep Learning TSC could provide, but illustrated a wide space for domain-specified explanation using SHAP. Two TSC algorithms, ROCKET and TS-KNN, showed the most accurate results and a better-defined explanation, as the most significant parts according to the SHAP interpretation were those that were often looked at by clinicians where the a- and b-wave are formed. At the same time, WEASEL and TSF algorithms tended to overlook parts of the signals that were considered to be inconsequential parts (baseline) prior to the stimulus onset.

## Figures and Tables

**Figure 1 bioengineering-12-00951-f001:**
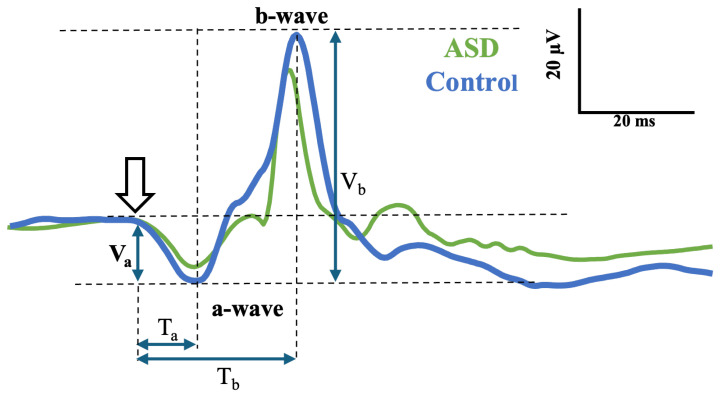
Illustration of a control (blue) and an ASD (green) light-adapted ERG signal. The amplitudes of the a- and b-wave are denoted as Va and Vb and the time to peak of these peaks as Ta and Tb, respectively.

**Figure 2 bioengineering-12-00951-f002:**
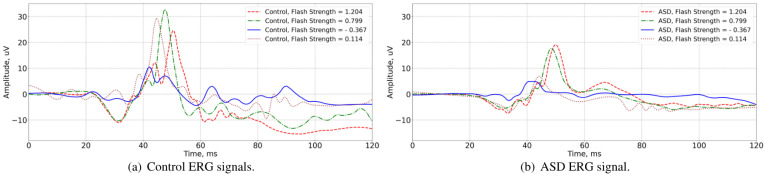
Illustration of a control (**a**) and ASD (**b**) ERG signal for different flash strengths (−0.367, 0.114, 0.799, and 1.204 log cd.s.m^−2^). Stimulus onset was at t=20 ms. Note the reduced positive b-wave amplitude in the 40 to 60 ms region for the ASD group.

**Figure 3 bioengineering-12-00951-f003:**
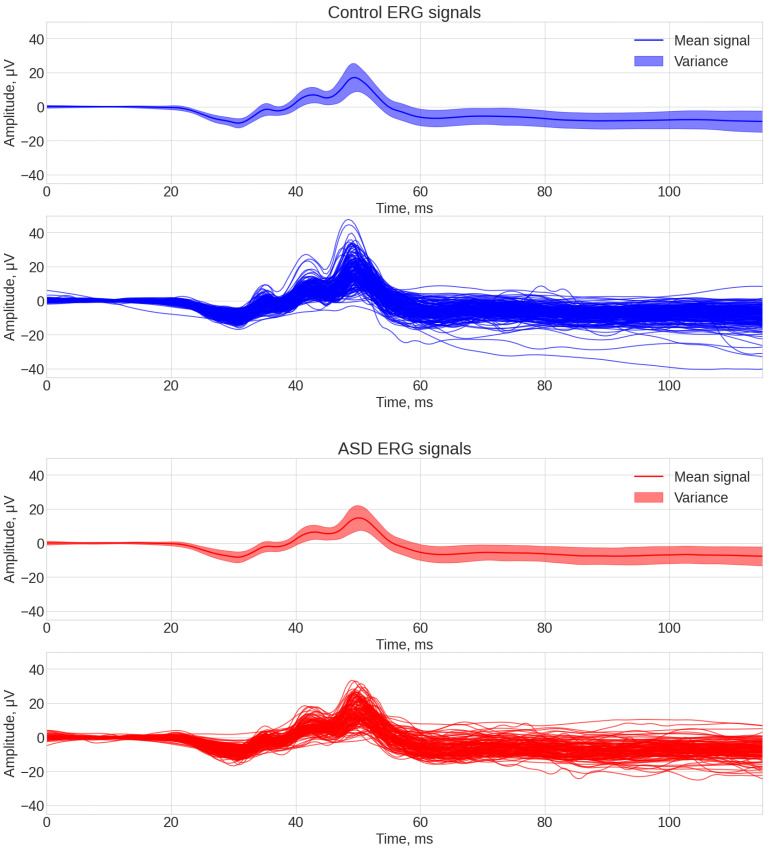
ERG signals for the control (**top**) and ASD (**bottom**) participants. Plots show the mean and standard deviation of the signals (**upper**) and individual waveforms (**lower**) for the 1.204 cd.s.m^−2^ flash strength.

**Figure 4 bioengineering-12-00951-f004:**
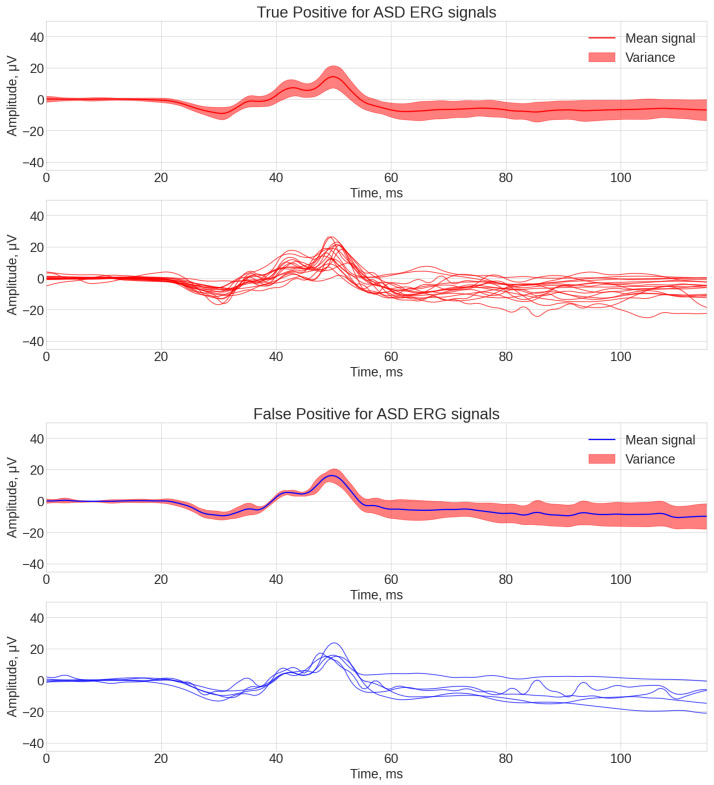
ROCKET classification examples of ERG waveforms assigned correctly or incorrectly to each class of ASD or control: true positive for ASD (**top**), false positive for ASD (**bottom**).

**Figure 5 bioengineering-12-00951-f005:**
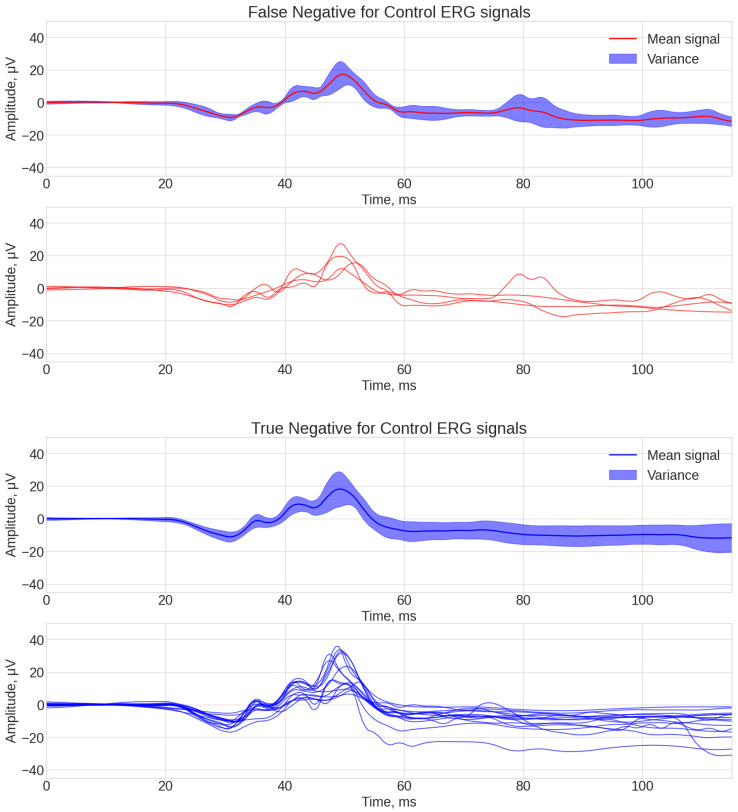
ROCKET classification examples of ERG waveforms assigned correctly or incorrectly to each class of ASD or control: false negative for control (**top**), true negative for control (**bottom**).

**Table 1 bioengineering-12-00951-t001:** Summary results for TSC models with the best hyperparameters on the test data.

TSC	F1 Control	F1 ASD	Balanced Accuracy
WEASEL	0.56±0.11	0.60±0.04	0.60±0.05
TSF	0.58±0.08	0.58±0.08	0.63±0.07
TS-KNN	0.60±0.08	0.56±0.07	0.59±0.07
ROCKET	0.66±0.10	0.63±0.07	0.66±0.10

## Data Availability

Constable, Paul; Marmolejo-Ramos, Fernando; Thompson, Dorothy; Brabec, Marek (2022). Electroretinogram raw waveforms for control and autism. Flinders University. Dataset. https://doi.org/10.25451/flinders.21546210.v1.

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
