# Peer review of "Time Series Classification of Autism Spectrum Disorder Using the Light-Adapted Electroretinogram"

_bioengineering, 2025, doi:10.3390/bioengineering12090951_

Round 1

Reviewer 1 Report

Comments and Suggestions for Authors

Submitted manuscript is devoted to the attempt to use the light adapted electroretinogram for classification of autism spectrum disorder. Although, the subject is interesting and somehow aligned with the journal keywords, neither title nor Introduction provide clear and convincingly formulated goal of the study. Essential part of the work is rather short and description of content at the end of the Introduction is not necessary. Instead, at the end of the Introduction authors must provide the aim of the work, which is absent in the present version.

Appendix of the manuscript contains a lot of figures without analysis. Essential part of the analysis can be given in Appendix; however, it would be better to place it as a supplementary part. In addition, the content of each figure must be clear from the figure captions without need to consult the main text  – here it is not a case.

All abbreviations must be defined at their first appearance. For instance, ROCKET appears simply as it is. One can guess that in biomedicine, "ROCKET" can refer to a cloud-based platform for managing and analyzing biomedical data, a clinical trial program for a drug, or even a novel drug delivery system inspired by rocket design. 

All symbols appearing for each figure must be at least the size of the figure caption font. In many cases they are too small for comfortable reading even for person with normal visual parameters.

Conclusions are not really supported by the careful analysis. The declaration “The most popular TSC techniques: WEASEL, TS-KNN, ROCKET and TSF were tested 383 for an ERG signals classification problem for the first time” is rather indication of the research direction as opposed to conclusion supported by detailed analysis. It would be very important to compare the data obtained from the light adapted electroretinography with real clinical data for a number of patients whose state was evaluated by other clinical studies.

Conclusions contain parts which should be moved to a Discussion section.

Author Response

Dear Reviewer,

We would like to thank you for your consideration of our manuscript. We have attached our revised version of the manuscript, along with a detailed response to all your comments.

Our revisions have addressed all your concerns, and we hope that the updated manuscript is now suitable for publication.

Best regards, 
On behalf of all authors

Reviewer 2 Report

Comments and Suggestions for Authors

Good Day,

The present manuscript has, as main goal, the classification of autism spectrum disorder by using the light adapted electroretinogram, a subject of great interest nowadays.

The Abstract mentions the aim of the research and, shortly, some findings of those studies. The cited References are relevant to the field. The English is fine and clear, and the Ithenticate report is remarcably low.

Please check also on the entire manuscript, that all the used abbreviations are explained at least first time when they appear (if it is the case), as for example ERG, ASD, OPs, SHAP (which are described).

The Introduction is relevant and makes an overview of the current state-of-the-art of the field, but it could be improved by adding more information and by adding more citations (especially from the last decade).

The materials and methods are described in details. It is not clear why Figs. 2a and 2b are presented at the methods, and not at the results section, where it could belong. Please explain that, or if necessarry, move those figs. at results.

The results and discussion are clearly presented. It would be interesting also to discuss the potential of using ERG not only for ASD population who already developed autism simptoms, but also for the population which could / will develop it in the future (practically it is present from the birth, but still not developed). It could this method predict such condition or there are other (maybe better) methods to do this? There are children who developed the condition later in life. In general, in the first years, there are no significant and visible simptoms, so maybe a method like ERG could already detect to risk for autism disorder even without significant simptoms.

The Conclusions are sustained by the presented and the discussed methods and results.

After performing the revision in agreement with the above mentioned suggestions, answering the questions, adding more information and more citations, and also more discussions and explanations (including all the used abbreviations if it is the case), the manuscript could be accepted for publication.

Best regards!

Author Response

Dear Reviewer,

We would like to thank you for your consideration of our manuscript.

Please see the attachment of our revised version of the manuscript, along with a detailed response to all your comments.

Our revisions have addressed all your concerns, and we hope that the updated manuscript is now suitable for publication.

Best regards, 
On behalf of all authors

Round 2

Reviewer 1 Report

Comments and Suggestions for Authors

Submitted revised manuscript is devoted to the attempt of usage the light adapted electroretinogram for classification of autism spectrum disorder. Although, the subject is interesting and somehow aligned with the journal keywords, neither title nor Introduction provide clear and convincingly formulated goal of the study. Essential part of the work is rather short and description of content at the end of the Introduction is not necessary. Instead, at the end of the Introduction authors must provide the aim of the work.

Appendix of the manuscript contains a lot of figures without analysis. Essential part of the analysis can be given in Appendix; however, it is necessary to place it as a supplementary part.

Conclusions are not really supported by the careful analysis. It would be very important to compare the data obtained from the light adapted electroretinography with real clinical data for a number of patients whose state was evaluated by other clinical studies.

Authors ignore the major part of the questions and did not make changes in the text.

Author Response

Dear Reviewer,

We would like to thank you for your reconsideration of the paper. We have attached our revised versionalong with a detailed response to all your comments. Please see the attachment.

Our revisions have addressed all your concerns, and we hope that the updated manuscript is now suitable for publication.

Best regards, 
On behalf of all authors

Round 3

Reviewer 1 Report

Comments and Suggestions for Authors

Work was improved up to the level suffisient for publication in this journal.